# Bacterial Drug Delivery Systems for Cancer Therapy: “Why” and “How”

**DOI:** 10.3390/pharmaceutics15092214

**Published:** 2023-08-27

**Authors:** Xiangcheng Zhao, Nuli Xie, Hailong Zhang, Wenhu Zhou, Jinsong Ding

**Affiliations:** 1Xiangya School of Pharmaceutical Science, Central South University, Changsha 410006, China; 217211037@csu.edu.cn (X.Z.); nulixie@csu.edu.cn (N.X.); zhl@king-eagle.cn (H.Z.); 2Changsha Jingyi Pharmaceutical Technology Co., Ltd., Changsha 410006, China

**Keywords:** cancer therapy, bacteria, drug carrier, drug-loading strategies

## Abstract

Cancer is one of the major diseases that endanger human health. However, the use of anticancer drugs is accompanied by a series of side effects. Suitable drug delivery systems can reduce the toxic side effects of drugs and enhance the bioavailability of drugs, among which targeted drug delivery systems are the main development direction of anticancer drug delivery systems. Bacteria is a novel drug delivery system that has shown great potential in cancer therapy because of its tumor-targeting, oncolytic, and immunomodulatory properties. In this review, we systematically describe the reasons why bacteria are suitable carriers of anticancer drugs and the mechanisms by which these advantages arise. Secondly, we outline strategies on how to load drugs onto bacterial carriers. These drug-loading strategies include surface modification and internal modification of bacteria. We focus on the drug-loading strategy because appropriate strategies play a key role in ensuring the stability of the delivery system and improving drug efficacy. Lastly, we also describe the current state of bacterial clinical trials and discuss current challenges. This review summarizes the advantages and various drug-loading strategies of bacteria for cancer therapy and will contribute to the development of bacterial drug delivery systems.

## 1. Introduction

Cancer is one of the major diseases that endanger human life and health. According to a report by the International Agency for Research on Cancer, there were about 19.3 million new cases and 10 million deaths of cancer patients worldwide in 2020 [1]. However, the current treatment methods, such as chemotherapy and radiotherapy, are accompanied by serious side effects, such as gastrointestinal toxicity, bone marrow suppression, alopecia, and so on [2]. The occurrence of these side effects is mainly related to the non-targeted distribution of drugs. Therefore, the development of effective tumor-targeted delivery systems is one of the main development directions of anticancer drug delivery systems. Nowadays, widely studied drug delivery systems, such as liposomes, micelles, and nanoparticles (NPs), have been shown to improve the anticancer effects of drugs, but still have some shortcomings, including low biocompatibility, high off-target effects, and rapid clearance in the blood. Dai et al. quantified the cancer cell targeting efficiency of nanoparticles in solid tumors, showing that only 0.7% of intravenous NPs were delivered to solid tumors, and only 0.0014% of NPs were delivered to cancer cells [3]. Most NPs are either captured in the extracellular matrix or absorbed by perivascular tumor-associated macrophages. Therefore, it is urgent to develop new and effective drug delivery systems to overcome the existing limitations.

In recent years, bacterial-based biocarriers have received widespread attention in the field of cancer therapy due to their obvious tumor targeting, oncolytic, and immunomodulatory properties. The use of bacteria in cancer treatment can be traced back to the 19th century. Coley et al. successfully achieved tumor reduction by injecting Streptococcus into cancer patients [4]. Currently, most of the bacteria used in cancer therapy are obligate and facultative anaerobic bacteria, such as *Salmonella typhimurium* (*S. typhimurium*), *Listeria*, *Escherichia coli* (*E. coli*), *Bifidobacterium*, *Clostridium*, and so on [5,6,7,8,9]. Some self-mineralizing and magneto-aerotactic bacteria have also been tried for oncology treatment [10,11]. Researchers have tried to carry anticancer substances, such as drug molecules, immune factors, and nucleic acids, on the surface of bacteria through chemical bonds or electrostatic adsorption, or inside bacteria through incubation or gene editing. Drug-loaded bacteria will release drugs after reaching the tumor site, thereby achieving tumor-targeted drug delivery. Under the combined action of drugs and bacteria, cancer cells undergo autophagy and apoptosis, while the body’s immune system will be activated, to achieve multiple pathways to inhibit tumor growth and diffusion [12,13]. Unlike general delivery systems that can only be passively transported to the surface of the tumor, flagellated bacteria can also move autonomously, which can make up for the lack of permeability of tumor tissue in existing drug delivery systems.

It is worth noting that with the development of gene editing technology, the safety and feasibility of modern bacterial therapy have been greatly improved. For example, the knockout of toxic genes can significantly reduce the toxicity of engineered bacteria compared with wild strains [14,15], and the knockout of nutrient-producing genes can make auxotrophic engineered bacteria colonize far more in tumors than in normal tissues [16,17]. Based on the potential of bacterial carriers in cancer therapy, this review will introduce the advantages of bacterial carriers and various drug-loading strategies of bacteria, as well as discuss the existing limitations and future development trends of bacterial carriers.

## 2. The Advantages and Mechanisms of Bacteria for Cancer Therapy

### 2.1. Tumor-Targeting of Bacterial Carriers

As a new type of anticancer drug carrier, bacteria will preferentially accumulate in tumors after entering the human body. Compared with healthy tissues, the accumulation of bacteria in tumor tissues is more than 1000 times higher [15]. Unlike liposomes, micelles, and NPs which can only passively target tumors by bloodstream transport, the bacteria have both passive and active targeting mechanisms and can actively penetrate deep into the tumor tissue [18,19]. The generation of the targeting property can be related to the suitable tumor microenvironment (TME) and the properties of bacteria. In addition, various modification methods can further improve the tumor-targeting ability of bacteria (Figure 1).

#### 2.1.1. Suitable TME for Bacterial Survival

The external reasons for the tumor targeting of bacteria can be roughly divided into three points. Firstly, the hypoxia TME attracts obligate and facultative anaerobic bacteria to tumor tissue. Due to the rapid proliferation of tumor cells and incomplete vascular development, the supply of oxygen in solid tumors is insufficient, which eventually leads to the existence of hypoxic areas in tumors [20]. This is a necessary condition for the survival and reproduction of obligate and facultative anaerobic bacteria in tumors. Moreover, with further penetration of the tumor tissue, hypoxia tends to be more severe. This is very unfavorable for conventional treatment but is more conducive for bacterial tropism and depth of penetration [21,22]. Secondly, the rich nutrients in the tumor tissue are another important reason for attracting bacterial colonization. Kasinskas et al. demonstrated that aspartic acid, serine, ribose, and galactose in the tumor can help *Salmonella* chemotaxis [23]. Song et al. demonstrated that clusterin is one of the key biochemical molecules in *E. coli* chemotaxis of lung cancer cells [24]. Lastly, the Immunosuppressive properties of tumors benefit bacterial colonization [25]. Bacteria that colonize tumor sites in the early stages can avoid clearance by the immune system, while bacteria that travel to normal tissues will be cleared by the immune system. Therefore, the concentration of bacteria in the tumor site is often much higher than in other sites, and the bacteria do not colonize the non-tumor-related hypoxic or inflammatory lesions [26,27,28,29].

#### 2.1.2. The Chemotaxis Properties of Bacteria

Some of the properties of bacteria can also help them to colonize tumors. It has been proposed that some chemical-specific receptors on the bacteria may sense chemicals secreted by cancer cells [19,23]. Kasinskas et al. proved that chemical receptors have an important effect on the tumor tropism of bacteria by knocking out different chemical receptors on the surface of *Salmonella* [23]. More interestingly, the accumulation of bacteria in different parts of the tumor can be controlled by selectively eliminating chemical receptor genes. In addition, some bacteria have special tendency properties. For example, magnetotactic bacteria have a tendency to a specific magnetic field [30]. By giving an auxiliary magnetic field in vitro, it can induce magneto-aerotactic bacteria to migrate to the tumor site. 

There are two opposing views on the relationship between bacterial motility and tumor tropism. One side believes that motility is essential for the effective distribution and accumulation of bacteria in tumor tissues. Studies by Kasinskas et al. and Toley et al. both used in vitro models to demonstrate that motility is essential for the effective distribution of bacteria in tumors [23,31]. On the contrary, Stritzker et al. found that chemotaxis and motility did not seem to help the bacteria colonize the tumor by intravenously injecting *E. coli* and *Salmonella* strains into the BALB/c mice with 4T1 breast cancer [32]. However, this study only analyzed the distribution of bacteria within the tumor at 48h after injection. Later, Ganai et al. conducted a longer experimental study on the same animal model [33]. Interestingly, the results showed that the bacteria began to migrate from the tumor edge to the central core after 48 h, and began to move closer to the tumor transition zone after 96 h. This may indicate that the experiment of Stritzker et al. still has some flaws and, if they could do longer tests, they may give a different conclusion.

It is undeniable that the movement of bacteria appears to be very insignificant compared to the speed of blood flow [34]. Therefore, bacteria entering the animal’s body are mainly transported passively by blood flow to the tumor site. But, upon reaching the tumor, the motility of the bacteria plays a crucial role in helping the bacteria penetrate deeper into the core of the tumor. When bacteria penetrate the tumor tissue, they will multiply in the tumor. Combined with the previously described immune clearance phase in normal tissues, this results in a much larger number of bacteria at the tumor site than in normal tissues.

#### 2.1.3. Tumor-Targeting Modifications of Bacterial Carriers

Gene editing and surface modification can further improve bacteria’s tumor-targeting ability. With the use of gene editing technology, it is possible to design modified auxotrophic bacteria corresponding to some specific purines, amino acids, and other nutrients within the tumor. Zhao et al. designed a leucine–arginine auxotroph *S. typhimurium* [35]. The engineered bacteria can only survive in tumors but not in normal tissues. In terms of surface modification, modifying tumor-homing peptides or tumor antibodies on the surface of bacteria can lead to better tumor-targeting ability. Park et al. improved the tumor tropism of bacteria by modifying an arginine–glycine–aspartate peptide on the outer membrane protein of *S. typhimurium* [36]. Massa et al. demonstrated that the tumor specificity of *Salmonella* can be significantly improved by surface-expressed antibodies against tumor-associated antigens [37].

### 2.2. Immunomodulatory Effects of Bacterial Carriers

Cancer patients cannot clear cancer cells through their immune system, because cancer cells can develop multiple pathways to avoid being cleared by the immune system [38]. It has been found that bacteria inside the tumor can alter the immunosuppressive TME and stimulate the host immune system, thus enhancing the body’s immune system to clear the cancer cells (Figure 1).

#### 2.2.1. Weak Antitumor Immunity

The reasons for the inability of the anticancer immune response to eliminate tumors can be broadly classified as follows:Tumor cells themselves have developed specialized mechanisms to suppress immune responses, including downregulation of tumor antigen and major histocompatibility complex (MHC) class I expression [39,40], high expression of programmed death receptor-ligand 1 (PD-L1) to prevent T cell activation [41], and expression of various immunosuppressive cytokines and chemokines by themselves or induced tumor-infiltrating immune cells [38,42].The presence of a large number of immunosuppressive cells within the tumor, such as tumor-associated macrophages (TAMs), regulatory T cells (Tregs), and myeloid-derived suppressor cells (MDSCs), can significantly inhibit the infiltration of cytotoxic T lymphocytes [43,44,45].The patient’s immune function is so weak that the growth rate of tumor cells exceeds the clearance rate of the immune system [46]. The combination of these factors makes it difficult for the body to rely on its immune system to remove tumor cells, thus allowing the tumor to grow and spread.

#### 2.2.2. Bacteria Activate the Immune System

Bacteria at the tumor site can stimulate the immune system to produce a series of anticancer immune responses is another advantage of bacterial carriers in cancer therapy. This is because bacteria carry a large number of antigens such as lipopolysaccharide and flagellin, which can bind to toll-like receptors and, thus, trigger a series of cellular signaling events [47,48,49]. The mechanisms by which bacteria activate anticancer immunity in the body are complex and interact with each other. For the sake of description, we roughly divide these mechanisms into two parts: promotion of anticancer immune response and reduction of tumor immune escape (Figure 2).

In enhancing the anticancer immune response, the recruitment of immune cells to reach the tumor site is the most fundamental step. Several studies have demonstrated that bacteria at the tumor site can induce a large number of immune cells to accumulate toward the tumor, including macrophages, NK cells, B cells, CD4+ T cells, CD8+ T cells, and so on [50,51,52]. When a large number of immune cells reach the tumor site, a series of anticancer immune responses are triggered. Secondly, bacteria can increase the number of TANs and, thus, enhance the innate immune response [53]. However, neutrophils show two sides in the development of tumors [54,55,56]. TAN1 exhibits anticancer activity through direct cytotoxic effects and activation of adaptive immune responses, while TAN2 exhibits tumor-promoting activity by promoting tumor cell proliferation, migration, and invasion, stimulating angiogenesis and mediating immunosuppression. Furthermore, bacteria promote gap junctions between DCs and cancer cells by upregulating connexin 43 to enhance the ability of dendritic cells to present tumor antigens [57,58]. This process leads to the secretion of large amounts of the proinflammatory cytokine IL-1β by DCs and subsequent activation of CD8+ T cells. Lastly, bacteria were found to activate the NF-κB pathway, which in turn upregulates the expression of various immunostimulatory and chemokines, including IL-6, IL-13, IL-17, IL-1β, G-CSF, GM-CSF, MIP-1α, TNF-α, and IFN-γ [59,60,61,62]. 

Reducing tumor escape is another important means to activate antitumor immunity. As mentioned above, tumor immune escape is associated with a large number of immunosuppressive cells, including TAMs, Treg, and MDSCs, which protect tumor cells in various ways to treat the activation of cytotoxic T cells [62]. Among them, TAMs are usually divided into two opposite subtypes, including M1 macrophages that exert antitumor activity and M2 macrophages that inhibit T-cell-mediated antitumor immune response. Both M1 and M2 macrophages have a high degree of plasticity, so they can be transformed into each other when the tumor microenvironment changes or therapeutic interventions [63]. It was found that *S. typhimurium* expressing flagellin promoted the M2–M1 transition of macrophages and increased the level of nitric oxide in tumors through the synergistic effect with TLR4 and TLR5 signaling pathways [64]. In addition, flagellin stimulates NK cells to produce IFN-γ, which can also promote the conversion of M2 to M1 [65]. Treg cells and MDSCs are the other two important immunosuppressive cells. Treg cells can inhibit the costimulatory signal of CD80 and CD86 expressed by dendritic cells, consume IL-2, secrete inhibitory cytokines, and directly kill effector T cells [66]. MDSCs play an important role in tumor angiogenesis, drug resistance and tumor metastasis [67]. Studies have found that some bacteria can effectively reduce Treg and DMSCs in tumor tissues. For example, attenuated *Salmonella* vaccine has been shown to induce immunosuppressive myeloid-derived suppressor cells to transform into TNF-αsecreting cells with neutrophil characteristics and significantly reduce Treg cells [68,69,70]. Secondly, *Listeria* was found to be delivered to metastatic and primary tumors by infecting DMSCs, significantly reduce the number of MDSC in blood and primary tumors, and transform the remaining MDSC subsets into an immunostimulatory phenotype that produces IL-12 [71]. In addition, bacteria downregulate the expression of immunosuppressive factors such as IL-4, ARG-1, and TGF-β [62,72]. 

### 2.3. Oncolysis of Bacterial Carriers

In addition to activating the immune system to kill cancer cells, bacteria colonizing tumors also have a certain anticancer effect, which can help drugs better play their anticancer role. This oncolytic activity can be achieved through a variety of pathways, including the induction of tumor cell death, inhibition of tumor angiogenesis, inhibition of tumor metastasis, and reduction of tumor drug resistance (Figure 1).

#### 2.3.1. Induction of Tumor Cell Death

Bacteria induce cancer cell death through multiple pathways, including the induction of apoptosis, the release of bacterial toxins, and competition for nutrients. For example, *Listeria* can induce tumor cell apoptosis by activating nicotinamide adenine dinucleotide phosphate (NADPH) oxidase and increasing intracellular calcium levels to increase ROS levels in tumors [73]. *Salmonella* induces autophagy and caspase-mediated apoptosis in tumor cells by downregulating the AKT/mTOR pathway [74], and metabolizes nitrate to nitrite via nitrate reductase and further converts it to nitric oxide in tumors to induce tumor cell apoptosis [75]. Secondly, bacteria can release toxins to kill tumor cells. For example, Colicins have anticancer activity against breast cancer, colon cancer, bone cancer, and other human tumor cell lines [76]. However, bacterial toxins are a double-edged sword. Strong toxicity may lead to damage to other normal tissues while showing good tumor killing. Therefore, it is necessary to select the appropriate intensity of toxicity as well as the dose. Lastly, the growth and reproduction of bacteria at the tumor site can consume a large number of nutrients, and by competing with tumor cells for nutrients it can promote apoptosis of some tumor cells [77].

#### 2.3.2. Inhibition of Tumor Angiogenesis

Tumor growth is closely linked to tumor angiogenesis. The rapid growth of tumor cells requires a large amount of oxygen and nutrients; therefore, a large number of blood vessels are generated inside the tumor to transport the materials required by the tumor cells. If angiogenesis fails, the lack of oxygen and nutrients limits the tumor diameter to 2–3 mm [78]. The hypoxic TME can stimulate the tumor to produce some adaptive responses. At this time, the tumor cells will activate the hypoxia-inducible factor (HIF) signal and upregulate the expression of vascular endothelial growth factor (VEGF) and other proangiogenic factors, thereby increasing tumor angiogenesis [79,80]. *Salmonella* was found to downregulate HIF-1α expression via downregulation of the AKT/mTOR pathway, which in turn inhibited tumor VEGF expression and angiogenic signaling [81].

#### 2.3.3. Inhibition of Tumor Metastasis

The metastasis of malignant tumors is frequently the main cause of tumor treatment failure, and the degradation of extracellular matrix caused by matrix metalloproteinase-9 (MMP-9) plays a key role in this process. Cancer cells can break through the physical barrier of the extracellular matrix by influencing host cells to secrete MMP-9 to degrade multiple collagen proteins in the basement membrane to metastasize and invade other tissues [82]. Tsao et al. demonstrated in a mouse model of melanoma and lung cancer that *Salmonella* inhibited MMP-9 expression by downregulating the AKT/mTOR signaling pathway, resulting in the inhibition of tumor cell migration and reduction of nodule production in vivo [83].

#### 2.3.4. Reduction of Tumor Drug Resistance

Reducing the drug resistance of tumor cells is also one of how bacteria achieve oncolysis. p-glycoprotein (P-gp), also called multidrug resistance protein, is a protein that pumps certain intracellular chemicals out of the cell and reduces their concentration in the cell for cell protection [84]. The P-gp is also distributed on the surface of tumor cells, which allows the chemotherapeutic drugs inside the tumor cells to be excreted extracellularly, thus causing the tumor cells to develop drug resistance. In mouse melanoma and mammary tumor models, *Salmonella* reduced the expression of P-gp by inhibiting the expression levels of phosph-protein kinase B, phosph-mammalian targets of rapamycin, and phosphate-p70 ribosomal s6 kinase in tumor cells., thus improving the sensitivity of tumors to chemotherapy [85].

## 3. Drug-Loading Strategies of Bacterial Carriers

### 3.1. Drug Loading on the Surface of Bacteria

The drug-loading strategies on the surface of bacteria can be divided into chemical bonding, linker grafting, and physical adsorption. Chemical bonding is to use the chemical groups on the surface of bacteria to form covalent bonds with drugs to achieve connection. Linker grafting is to use the noncovalent force between ligand and receptor to realize the loading of goods. Physical adsorption is mostly realized by the Coulomb force between the surface of bacteria and drugs. Compared with different surface drug loading methods, the connection structure of chemical bonding and linker grafting is relatively stable, and it is not easy to change the structure and active function in vivo. Physical adsorption has the advantages of simple operation and less damage to bacteria, but the stability of its connection structure is relatively weak, and drug shedding may occur (Figure 3 and Table 1).

#### 3.1.1. Chemical Bonding

The surface of bacteria is mainly composed of peptidoglycan, lipopolysaccharide, and polypeptide, so there are a large number of chemical groups such as amine, sulfhydryl, and hydroxyl groups on the surface [86,87]. The cargo can react with groups to form covalent bonds to the surface of the bacteria and the most typical reaction is the amidation reaction between amino and carboxyl groups [88]. In addition, in situ polymerization based on biomineralization and oxidative self-polymerization is also a covalent binding method commonly used for bacterial surface modification.

Chen et al. loaded zeolitic imidazole frameworks-90 (ZIF-90) encapsulated with the photosensitizer methylene blue (MB) onto the surface of self-mineralizing photothermal bacteria, enabling the combination of bacteriotherapy and tumor photothermal therapy (Figure 4A) [10]. The MB-encapsulated ZIF-90 (ZIF-90/MB) formed an acid-sensitive imine covalent bond with the amino group on the surface of bacteria, thus enabling drug loading. Fan et al. developed an integrated bioreactor based on engineered bacteria to treat tumors by Fenton-like reactions and local production of H_2_O_2_ [89]. They used N-acetylmuramic acid in the cell wall of Gram-negative bacteria to chemically modify aminated magnetic Fe_3_O_4_ NPs to the surface of engineered *E. coli* that highly expresses respiratory chain enzyme II. Liu et al. designed an *E. coli* carrying immunoreactive polydopamine nanoparticles, which carry αPD1 and S1 proteins that can produce long-term stimulation to immune cells [90]. Based on the principle that dopamine can react with amino acid residues on the surface of bacteria to form covalent bonds, researchers modified nanoparticles on the surface of bacteria by a simple one-step method.

#### 3.1.2. Linker Grafting

The ligand can form a stable connection with the receptor through various non-covalent bonding forces, such as hydrogen bonds, van der Waals forces, charge forces, and hydrophobic interactions. Biotin–streptavidin linkage and antigen–antibody interactions are currently the most commonly used strategies. By modifying antibodies or biotin on the surface of cargo or bacteria, the carrying mode of “drug-linker-bacteria” can be realized.

Suh et al. achieved the loading of NPs on the surface of bacteria utilizing the bioconjugation method based on biotin-streptavidin interaction (Figure 4B) [91]. Specifically, streptavidin was modified on the NP’s surface, while the biotin antibody was modified on the surface of *Salmonella*. Noncovalent bond coupling was achieved by the interaction between streptavidin and biotin. The results showed that the NPs carried by bacteria could effectively overcome the barrier of extravascular transport and greatly improved the retention and distribution of NPs in tumors. In addition, Wilber et al. successfully carried radioactive sources onto the surface of *Listeria* bacteria by incubating them with bacterial antibodies containing 188-rhenium (^188^Re) [92]. Radioactive bacteria accumulate in pancreatic cancer tumors in the body, achieving local radioactivity and killing tumor cells without serious side effects.

#### 3.1.3. Physical Adsorption

The outer membrane of bacteria contains a lot of teichoic acids, peptidoglycan, and protein, which makes the isoelectric point of the bacterial surface 2~5 [93]. In general, the pH of bacterial culture and test environment is 6~7.5, which is higher than the isoelectric point of bacteria, so the surface of bacteria often presents a negative potential state. Therefore, positively charged substances can be adsorbed on the surface of bacteria by Coulomb force to achieve drug assembly.

Substances with a positive charge can be adsorbed on the surface of bacteria, due to the negative charge of the bacterial surface. Zheng et al. assembled the nano-photocatalytic material Carbon nitride (C_3_N_4_) and *E. coli*, which can produce NO and, thus, obtain a biotic/abiotic hybrid for light-controlled NO generation (Figure 4C) [94]. Under the light, photoelectrons produced by C_3_N_4_ can be transferred to *E. coli* to promote the production of NO to make tumor cells dead. Hu et al. designed an effective oral DNA vaccine delivery system for cancer immunotherapy utilizing cross-linkedβ-cyclodextrin-PEI600 (CP)/DNA NPs and *Salmonella* [95]. The CP/DNA NPs are positively charged and can be adsorbed on the surface of attenuated *Salmonella* by electrostatic force. The protective nanoparticle coating can help bacteria effectively escape phagosomes, significantly enhance the acid resistance of bacteria in the gastrointestinal tract, and greatly promote the entry of bacteria into the blood circulation after oral administration.

**Figure 4 pharmaceutics-15-02214-f004:**
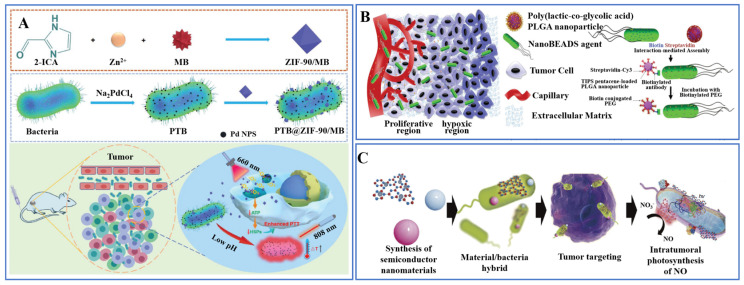
(**A**) Chemical bonding. ZIF-90/MB formed an acid-sensitive imine covalent bond with the amino group on the surface of bacteria [10]. Copyright © 2020 WILEY-VCH Verlag GmbH & Co. KGaA, Weinheim, Germany. (**B**) Linker grafting. Streptavidin-modified NP and biotin antibody-modified *Salmonella* form a “drug-linker-bacteria” connection structure [91]. Copyright © 2018, WILEY-VCH Verlag GmbH & Co. KGaA, Weinheim. (**C**) Physical adsorption. C_3_N_4_ adsorbs on the surface of *E. coli* by electrical charge and promotes NO production by bacteria under light [94]. Copyright © 2018, Springer Nature.

### 3.2. Drug Loading Inside Bacteria

In order to achieve drug loading inside bacteria, drugs can be directly loaded into bacteria cells, or loaded into bacteria in the form of nucleic acids by gene editing technology. The biggest difference between these two methods lies in the different ways of drug release. The direct loading method achieves a one-time release of drugs by lysing bacteria, while genetically modified bacteria can continuously produce anticancer substances (Figure 3 and Table 1).

#### 3.2.1. Direct Uptake

The pathways of bacterial uptake of external substances include passive diffusion and active transport, and drugs or nanoparticles can enter the bacteria through these pathways. When the bacteria reach the tumor region, the drug is released from inside the bacteria through some special responses, thus achieving the killing of cancer cells.

Sun et al. designed a ‘Trojan bacteria’ drug delivery system, which consists of bacteria and glucose polymer (GP)-conjugated and indocyanine green (ICG)-loaded silicon NPs (Figure 5A) [96]. During the co-culture of NPs and anaerobic bacteria, bacterial-specific transport proteins transported NPs into cells to achieve drug loading inside bacteria. Compared with NPs that struggle to enter the brain, the ‘Trojan bacteria’ drug delivery system could cross the blood–brain barrier, targeting and penetrating glioblastoma tissue. Under 808 nm laser irradiation, ICG released large amounts of thermal energy to lyse bacteria and destroy tumor cells, while the resulting bacterial debris also promoted anticancer immunity. Chandra et al. cultured *Listeria* in a phosphorus-free medium supplemented with 32-phosphorus (^32^P) as a nutrient to obtain radioactive *Listeria* [97]. ^32^P was derived from phosphoric acid, so the bacteria absorbed the radioactive material primarily through active transport, and the researchers speculated that this radioactive phosphorus would be present in various parts of the bacteria.

#### 3.2.2. Electroporation

Electroporation is a method to improve cell membrane permeability. By applying high voltage and short-term electrical pulses to cells, instantaneous pores are generated on the surface of the membrane, thereby promoting transmembrane transport of substances that are not easily permeable [98]. In order to ensure the biological activity of bacteria, a reversible electroporation technique is used to apply a short electric pulse to bacteria, thereby promoting the absorption of drugs or liposomes by bacteria. However, the electroporation method still causes damage to bacteria. Compared with the untreated control, the treated bacteria showed different degrees of decline in biological activity.

In the process of trying to carry doxorubicin (DOX) liposomes into *S. typhimurium*, Zoaby et al. compared the DOX liposome uptake rates of direct incubation and electroporation incubation [99]. The results showed that the uptake rate of bacteria to liposomes was less than 5% when the bacteria were directly incubated with liposomes for more than 4 h, while the uptake rate of bacteria to liposomes after electroporation reached 62%. Compared with the untreated control, the treated bacteria showed a decrease of about 20% in growth. Xie et al. used electroporation to load 5-fluorouracil (5-FU) and zoledronic acid (ZOL) into *E. coli* and modified Au nanorods on the surface of *E. coli* (Figure 5B) [100]. The modified bacteria could load 8.8% 5-FU and 10.5% ZOL, and their survival rate and movement speed were reduced to 87% and 88%. Under near-infrared illumination, Au nanorods generate a lot of heat to kill bacteria and tumor cells, and then the drug will be released from the dead bacteria to further kill tumor cells.

#### 3.2.3. Genetic Engineering

Gene editing of bacteria is to transfect DNA fragments expressing anticancer substances into bacteria in the form of plasmids and control bacteria to continuously produce anticancer-related substances such as cytotoxic substances, tumor antigens, immune factors, and other substances so that bacteria can produce long-term sustainable therapeutic effects inside the tumor.

Nguyen et al. designed an attenuated *S. typhimurium* that can express cytolysin A (ClyA) [101]. In order to avoid ClyA killing normal tissue cells, the bacterial ClyA gene also contains an L-arabinose promoter, which allows the ClyA gene to be activated only by L-arabinose. When the bacteria reach the tumor site, the promoter can be activated by injecting L-arabinose into the body, thus enabling the bacteria to continuously express ClyA to kill the tumor cells. In order to break the immune tolerance to the autoantigen of liver cancer cells, Chou et al. transformed the alpha-fetoprotein (AFP) gene plasmid into attenuated *S. typhimurium* [102]. The modified attenuated *S. typhimurium* could express AFP specific to liver cancer and then activate T cells to kill and clear the tumor. In the study of utilizing bacteria to produce cytokines to treat tumors, Yoon et al. designed a *Salmonella* carrying IFN-γ to treat melanoma [103]. Subcutaneous injection of modified *S. typhimurium* was effective in inhibiting tumor growth and prolonging survival in melanoma mice compared to unmodified phosphate-buffered saline.

More interestingly, unlike engineering bacteria that directly express antitumor substances, Din et al. designed a bacterial drug delivery system that can deliver drugs repeatedly, synchronously, and periodically utilizing gene technology, providing a new pulsed drug release strategy (Figure 5C) [104]. Signal molecule Acyl-homoserine lactone (AHL), AHL synthesis protein LuxI, and regulatory protein LuxR are three key factors to achieve the bacterial cycle. When population density is low, the AHL produced by the bacteria mainly diffuses outside the cell and does not accumulate within the bacteria; whereas, when population density increases, intracellular AHL accumulates and reaches a critical concentration that permits the production of a large number of lysin proteins, resulting in the lysis of most of the bacteria and the release of the drug. This periodic cycle may be able to combine the effects of circadian rhythms on host–microbe interactions to achieve more efficient bacterial drug delivery by regulating the frequency and amplitude of the population cycle.

**Figure 5 pharmaceutics-15-02214-f005:**
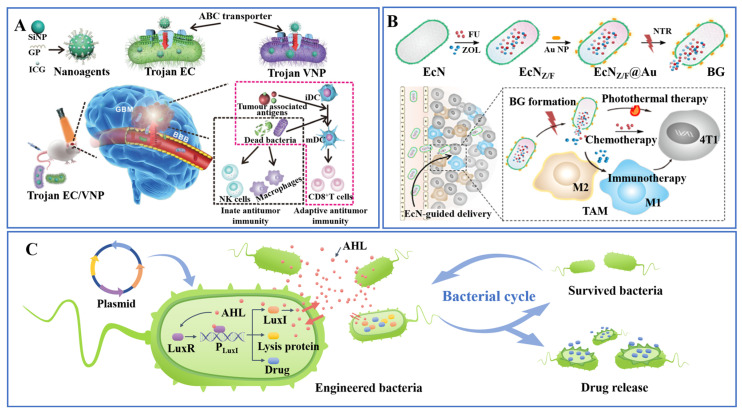
(**A**) Direct uptake. The bacteria take up the nanoparticles into the bacteria via special transporter proteins for drug loading [96]. Copyright © 2022, Springer Nature. (**B**) Electroporation incubation. EcN_Z/F_@Au are prepared by inoculation of FU and ZOL into EcN through electroporation and decoration of Au NRs on the EcN surface [100]. Copyright © 2021, Elsevier Ltd. (**C**) Genetic engineering. Bacteria transfected with the plasmid can continuously produce the drug and achieve pulsed release of the drug through the bacterial cycle [104].

**Table 1 pharmaceutics-15-02214-t001:** Delivery of anticancer drugs mediated by bacteria.

Drug-Loading Strategies	Bacterial Strain	Cargos	Type of Cancer	Ref.
Chemical bonding	*Shewanella oneidensis* MR-1	ZIF-90/MB NPs	Colon carcinoma	[10]
	*E. coli* MG1655	Fe_3_O_4_ NPs	Colon carcinoma	[89]
	*E. coli* Nissle 1917	PDA/PD1/S1 NPs	Colon carcinoma, Melanoma	[90]
	*S. typhimurium* VNP20009	Polydopamine	Melanoma	[105]
	*Magnetococcus marinus* MC-1	Carboxylated liposomes	Colon carcinoma	[11]
Linker grafting	*S. typhimurium* VNP20009	Poly (lactic-co-glycolic acid) NPs	Mammary cancer	[91]
	*Listeria* XFL-7	^188^Re	Pancreatic cancer	[92]
	*S. typhimurium* YS1646	DOX liposomes	Colon carcinoma	[106]
	*S. typhimurium* SHJ2037	paclitaxel liposome	Mammary cancer	[107]
Physical adsorption	*E. coli* MG1655	C_3_N_4_	Mammary cancer, Colon carcinoma	[94]
	*S. typhimurium*	CP/DNA NPs	Melanoma	[95]
	*E. coli* BL21	Bacterial toxin NPs	Renal carcinoma	[108]
Direct uptake	*S. typhimurium* VNP20009	GP-ICG-Si NPs	Glioblastoma	[96]
	*Listeria*	^32^P	Pancreatic cancer	[97]
	*S. typhimurium* Ty21a	Gold NPs	Colon carcinoma	[109]
Electroporation incubation	*S. typhimurium* LT2	DOX liposomes	Mammary cancer	[99]
	*E. coli* Nissle 1917	5-FU, ZOL, Au nanorods	Mammary cancer	[100]
Genetic engineering	*S. typhimurium* SHJ2037	ClyA	Colon carcinoma, Hepatoma	[101]
	*S. typhimurium*	AFP	Hepatoma	[102]
	*S. typhimurium* SHJ2037	IFN-γ	Melanoma	[103]
	*S. typhimurium* SHJ2037	Hemolysin E	Colon carcinoma	[104]
	*E. coli* DH5a	β-glucuronidase	Lung adenocarcinoma	[110]

## 4. Clinical Trials and Challenges

The BCG vaccine, which has been widely used in clinical practice, is a live vaccine made of attenuated bovine tuberculosis suspension, which indicates that it is feasible to carry out the clinical transformation of live bacteria [111]. Based on the better therapeutic effect of the bacteria in animal models, researchers began to try to conduct relevant clinical trials for human tumor treatment (Table 2). Some of these bacteria have shown good therapeutic efficacy when treated in combination with other drugs. For example, CRS-207 is a lister expressing mesothelin, and a series of clinical studies have been conducted with CRS-207 against pancreatic cancer, mesothelioma, lung cancer, and ovarian cancer, among other tumors [112,113,114,115,116]. In a phase I trial using Cyclophosphamide (Cy)/GVAX Pancreas vaccine in combination with CRS-207 for the treatment of metastatic pancreatic cancer, the results demonstrated that CRS-207 was safe and tolerable for cancer patients and prolonged the survival of pancreatic cancer patients [115]. However, in a subsequent phase II trial for patients with pancreatic cancer, the Cy/GVAX + CRS-207 group did not show a longer survival advantage than the chemotherapy group, but the results also showed that the survival rate with CRS-207 alone was close to the chemotherapy group [116]. These positive results have inspired researchers to further develop bacterial products for the treatment of tumors. 

However, bacteria that have shown good tumor suppressive properties in laboratory animal models do not always perform satisfactorily in the human body. Attenuated *S. typhimurium* VNP20009, an engineered bacteria lacking the purI and msbB genes, showed significant antitumor activity in mice [17]. But, in a clinical trial of intravenous VNP20009 for the treatment of patients with metastatic melanoma, VNP20009 failed to completely colonize tumors in patients despite administering high doses of VNP20009, in contrast to the results in a mouse tumor model [117]. This shows that there are interspecies differences in the efficacy of bacteria, which may be related to the clearance of bacteria by the immune system and the growth of bacteria in tumors. Taking larger doses of bacteria may address the problem of poor therapeutic efficacy, but larger doses of VNP20009 may trigger side effects caused by the release of various proinflammatory cytokines induced by other bacterial products such as flagellin. Therefore, simply increasing the dose of bacteria to improve the efficacy is not feasible. 

In addition to severe side effects at high doses and interspecies differences in efficacy, some deficiencies in bacteria deserve our attention: The amount of drug-carrying is low. Theoretically, when the surface is modified with too many drugs, the activity and motility of the bacteria will be limited. At the same time, the bacteria will not ingest the drugs without limit, because a large number of drugs entering the bacterial cells will also lead to the death of the bacteria themselves. As a result, the bacterial drug load is low for both surface loading and intracellular loading. Fortunately, genetically engineered bacteria can continuously produce targeting substances, which may compensate for this deficiency. Therefore, genetic modification may be the most promising drug delivery strategy.Bacteria are unable to eradicate tumors [118]. In the early stage, bacteria at the tumor site can accept the shelter of the immunosuppressive TME; but, with the ablation of the tumor, the immunosuppressive microenvironment is also changed. A large number of immune cells destroy bacteria and form an immune barrier, which gives cancer cells the chance to reappear [119]. In addition, resistance mutations in a very small number of cancer cells may also lead to tumor recurrence [120,121].The cost of using bacterial products may be high. In production, most of the modified bacteria have only been produced in small sizes in the laboratory, and there are still a large number of technical difficulties to be overcome for the large-scale production of stable modified bacterial preparations. In storage, in order to achieve long-term storage of bacteria, the commonly used measures are ultra-low-temperature freezing and freeze-drying preservation; freeze-drying technology is considered to be an advanced drying method for this kind of sensitive biological material [122]. In fact, the long-term storage of modified bacteria needs to consider not only the effect of the bacterial activity of the treatment means, but also the stability of the structure of the preparation. In addition, the storage of each bacterial preparation requires a separate space to avoid cross-contamination. From the point of view of production and storage, the use of bacterial preparations is currently costly, but these problems may also be overcome by more advanced technologies in the future.

The safety of bacterial therapy remains in question. As mentioned above, bacteria will proliferate in the tumor site of the patient, but some of them may still travel to normal tissues. Although the patient’s immune system will normally clear out the bacteria from normal tissues, it may still trigger side effects such as fever, hypotension, anemia, vomiting, diarrhea, nausea, and others [117]. Worse, if the patients are unable to clear the excess bacteria through their own immune system, sepsis can result and become life-threatening. Real-time monitoring of bacterial colonization in the patient is important for the safe administration of bacterial agents [123]. Several methods have been used to identify bacterial colonization within tumors, including bioluminescence, fluorescence, magnetic resonance imaging, and positron emission tomography [123,124,125]. In addition, it is important to choose the appropriate treatment regimen based on the patient’s actual condition, as the irrational use of bacterial agents may turn a “friend” into an “enemy”.

**Table 2 pharmaceutics-15-02214-t002:** Clinical trials of engineered bacteria for treating cancer.

Bacteria	Code	Phase	Type of Cancer	Ref.
*S. typhimurium*	VNP20009	Phase 1	Advanced or Metastatic Cancer	[117]
	VXM01	Phase 1, 2	Stage IV Pancreatic Cancer, Colorectal Cancer, Glioblastoma	[126,127]
	Saltikva	Phase 1, 2	Metastatic Pancreatic Cancer	[128]
	SGN1	Phase 1	Advanced Solid Tumor	[129]
	SS2017	Early Phase 1	Relapsed Neuroblastoma	[130]
	CVD908ssb	Phase 1	Multiple Myeloma	[131]
*Listeria*	CRS-207	Phase 1, 2	Malignant Epithelial Mesothelioma, Pancreas Adenocarcinoma, Non-small Cell Lung Cancer, etc.	[112,113,114,115,116]
	JNJ-64041809	Phase 1	Prostatic Neoplasms	[132]
	JNJ-64041757	Phase 1	Adenocarcinoma of Lung	[133]
	ADU-623	Phase 1	Astrocytic Tumors, Glioblastoma Multiforme, Anaplastic Astrocytoma, Brain Tumor	[134]
	ADXS-503	Phase 2	Lung Cancer, Non-Small Cell, Metastatic Squamous Cell Carcinoma, Metastatic Non-Squamous Cell Carcinoma	[135]
	ADXS11-001	Phase1, 2	Anal Cancer, Rectal Cancer, Squamous Cell Carcinoma, Small Cell Carcinoma, Head and Neck Cancer, etc.	[136,137]
*E. coli*	SYNB1891	Phase 1	Metastatic Solid Neoplasm, Lymphoma	[138]
*M. bovis*	VPM1002BC	Phase1, 2	Bladder Cancer	[139,140]
*C. novyi*	NT	Phase 1	Solid Tumor Malignancies	[141]

## 5. Summary and Prospect

As a new type of anticancer drug delivery system, bacteria can penetrate tumor tissues to achieve targeted drug delivery. In addition, it can stimulate a series of antitumor immune responses and inhibit tumor growth and migration. With the intervention of gene technology, bacterial carriers have stronger tumor targeting and higher safety. More interestingly, some engineered bacteria can spontaneously produce anticancer substances for continuous treatment and the progeny bacteria still have the ability to produce target substances. These biological characteristics make bacterial carriers show great potential in cancer treatment, and these are not available in other delivery systems such as liposomes, micelles, and nanoparticles.

It is worth mentioning that in addition to utilizing live bacteria as drug carriers, there is also the use of bacterial outer membrane vesicles [142], bacterial-derived microcells [143], and bacterial ghosts [144] for antitumor drug delivery. These bacterial-derived carriers have the surface properties of bacteria and, therefore, enable tumor-targeted delivery of drugs. Because these substances have no biological activity of bacteria, their safety is higher, but they also lose their autonomous movement performance, and the tissue penetration ability is greatly reduced.

In this review, we introduce the advantages and the main current drug-loading strategies of bacterial carriers in cancer therapy. For the drug-loading strategy, genetic modification may have greater development prospects in the future, because of the ability to continuously produce drugs. In addition, genetically modified bacteria have well-established production processes and can be stably produced on a large scale, compared to bacteria modified by other drug-loading strategies. More importantly, some genetically modified bacteria have entered the clinical stage. In the commercialization of bacterial-based drug delivery systems, it may be more feasible to develop bacterial formulations for veterinary use first, given the more pronounced antitumor effects in animals and the more relaxed regulatory regime for pharmaceuticals. There are precedents for this in other innovative technologies for vaccine development, such as DNA vaccines or recombinant viral vectors [145]. Currently, combining bacteria with other cancer therapies for tumor treatment has become a hot direction for basic research, but we must also think about the poor performance of bacteria in the clinical setting. Perhaps the current bacteria are not yet a mature and appropriate tool for cancer treatment, but we still expect that better bacteria will be designed and applied in the clinic to benefit a huge number of cancer patients.

## Figures and Tables

**Figure 1 pharmaceutics-15-02214-f001:**
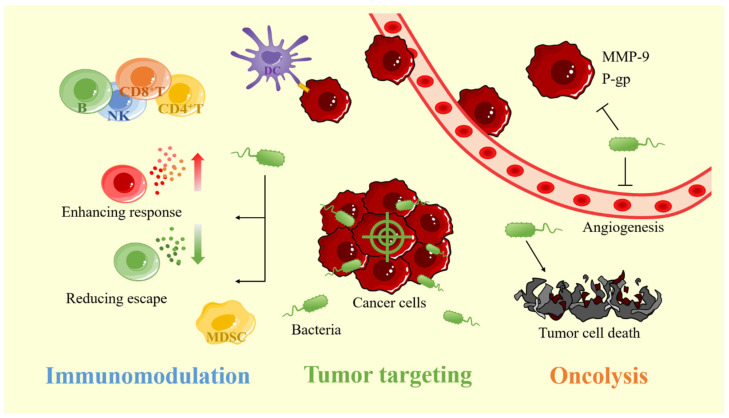
The advantages of bacterial carriers for cancer therapy include tumor-targeting, oncolytic and, immunomodulatory properties. MMP-9: matrix metalloproteinase-9; P-gp: p-glycoprotein.

**Figure 2 pharmaceutics-15-02214-f002:**
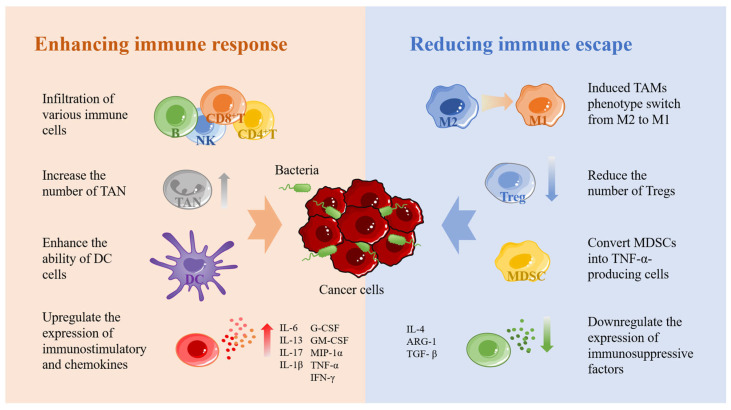
The immune stimulation mechanism of bacteria. Bacteria can recruit a large number of immune cells and produce a series of immune responses after colonization in tumors. NK: natural killer cells; TAN: tumor-associated neutrophils; DC: dendritic cell; M1: M1-like macrophage; M2: M2-like macrophage. Tregs: regulatory T cells; MDSC: myeloid-derived suppressor cells; IL-6: interleukin-6; IL-13: interleukin-13; IL-17: interleukin-17; IL-1β: interleukin-1β; G-CSF: granulocyte colony-stimulating factor; GM-CSF: granulocyte-macrophage colony-stimulating factor; MIP-1α: macrophage inflammatory protein-1α; TNF-α: tumor necrosis factor- α; IFN-γ: interferon-γ; IL-4: interleukin-4; ARG-1: arginase-1; TGF-β: transforming growth factor-β.

**Figure 3 pharmaceutics-15-02214-f003:**
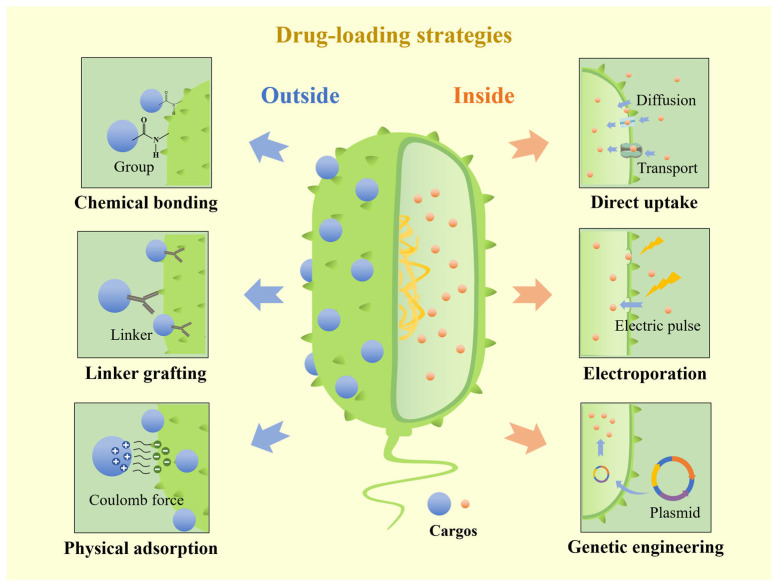
Various drug-loading strategies of bacterial carriers. Surface drug-loading strategies include chemical bonding, linker grafting, and physical adsorption. Inside drug-loading strategies include direct uptake, electroporation, and gene engineering.

## Data Availability

No new data were created.

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
