# Peer review of "Bacterial Drug Delivery Systems for Cancer Therapy: “Why” and “How”"

_pharmaceutics, 2023, doi:10.3390/pharmaceutics15092214_

Round 1

Reviewer 1 Report

In this review article, the authors have presented a comprehensive overview on the use of bacteria is a powerful tool for cancer therapy, while also focusing on the clinical trials in this regard. Honestly the review article is outstanding, and I enjoyed reading it, and I recommend its acceptance in the present form.

Reviewer 2 Report

This paper provides an in-depth and comprehensive literature review on the use of bacteria as drug-delivery scaffolds. The authors started with a proper explanation of the need for bacterial-based bio-carriers in cancer therapy and described 1) the advantages of using bacteria in cancer therapy, 2) bacteria as drug carriers, and 3) clinical trials and challenges. This reviewer acknowledges that it is an interesting topic for the novel strategy of cancer therapy. I believe that the reader will benefit from the following additional considerations.

1. The manuscript mainly contains the advantages of bacteria as drug delivery scaffolds and the parts that can be pointed out as disadvantages are a bit lack of description. In particular, safety issues and quality control aspects seem to be important for living drug delivery systems (even the yield of the scaffold preparation). Although this part is included in the challenge section, it would be good if it was described in more detail.

2. Some of the figures are hard to read because the text is too small (Figures 4, 5). This needs to be improved.

3. The paper would be enriched by including a section on commercialization efforts using these bacteria-based drug delivery systems.

4. If there is a bacterial drug delivery system that is more effective for different types of cancer, it would be good to add a description of that part.

Minor inappropriate word selection in the manuscript.

Reviewer 3 Report

This is an interesting review of how bacteria can be used to treat cancer and as drug delivery agents for that purpose. The paper is very well written, has good referencing, and useful figures. This is a topic I have not previously seen reviewed. A table of clinical trials is included. My only comment would be can the authors provide an example or two of how the 'specific response' works for release of a drug loaded inside bacteria.

Round 2

Reviewer 2 Report

I appreciate the great efforts that the authors have made in response to my questions and concerns. The revision clarifies all the points I raised and helps me (and hopefully readers) understand the current manuscript.